# Serological Evidence Supporting the Occurrence of *Ehrlichia chaffeensis* or a Closely Related Species in Brazilian Dogs

**DOI:** 10.3390/pathogens12081024

**Published:** 2023-08-09

**Authors:** Isis Indaiara Gonçalves Granjeiro Taques, Andreia Lima Tomé Melo, Mauricio Claudio Horta, Nathalia Assis Pereira, Daniel Moura Aguiar

**Affiliations:** 1Laboratory of Virology and Rickettsial Infections, Veterinary Hospital, Federal University of Mato Grosso, Av. Fernando Correa da Costa 2367, Cuiabá 78090-900, MT, Brazil; isis_indaiara@hotmail.com (I.I.G.G.T.); nathaliaassis89@gmail.com (N.A.P.); 2Faculty of Veterinary Medicine, University of Cuiabá, Cuiabá 78065-900, MT, Brazil; andreialtm@gmail.com; 3Laboratory of Parasitic Diseases, Federal University of São Francisco Valley, Petrolina 56304-917, PE, Brazil; mauricio.horta@univasf.edu.br; 4VetVida Veterinary Center of Diagnostic and Research, Cuiabá 78045-100, MT, Brazil

**Keywords:** ehrlichiosis, canine, serology, TRP32, ELISA, diagnosis

## Abstract

*Ehrlichia chaffeensis* is a tick-borne bacterium that causes human monocytotropic ehrlichiosis, an emerging life-threatening disease in humans transmitted by *Amblyomma americanum*. Although most studies have reported bacterial isolations and clinical cases in the United States, their occurrence is not restricted to North America. Some studies in the Southern Cone of South America have molecularly detected a close phylogenetic relative of *E. chaffeensis* in ticks and wild mammals. Even so, many gaps must be filled to confirm the presence of this agent in the region. To add new data on this issue, we report the first detection of specific anti-*E. chaffeensis* antibodies in dogs collected from all regions of Brazil. By means of IFA and ELISA with crude and specific antigens of *E. chaffeensis*, sera from 1134 dogs were analyzed. Serological analyses using ELISA showed nine (0.7%) seropositive dogs, with seven of them exhibiting IFA titers ranging from 160 to 5120. All regions of Brazil had at least one seropositive dog. Our results support the evidence for the occurrence of *E. chaffeensis* in South America. As dogs have a close relationship with humans, they can be used as an environmental sentinel for these infections because they can act as a bridge to human parasitism or infection with ehrlichial agents.

## 1. Introduction

*Ehrlichia chaffeensis* is a tick-borne bacterium that causes human monocytotropic ehrlichiosis (HME), an emerging life-threatening disease in humans that typically results in fever-like symptoms but may also be fatal if left untreated. In the United States, this infection has been increasing over the years, mainly in the states of the southeast and Midwest, as well as much of the east coast, where populations of white-tailed deer (*Odocoileus virginianus*), the main reservoir of *E. chaffeensis* and host of the lone-star tick (*Amblyomma americanum*), the vector of this disease, abound [1]. In addition to this species of deer, canines have also been characterized as susceptible to infection by *E. chaffeensis* [2]. For this reason, they can be considered environmental sentinels for this agent. In South America—specifically, Argentina and Brazil—*Ehrlichia* spp. that are similar or closely related to *E. chaffeensis* have been detected in ticks, namely, *Amblyomma parvum* [3], *Amblyomma tigrinum* [4], and *Rhipicephalus microplus,* as well as in marsh deer (*Blastocerus dichotomus*) [5] and non-human primates (*Mico melanurus*) [6]. A recent report provided evidence showing that non-human primates may be susceptible to *Ehrlichia* spp.

The diagnosis of canine infection by *E. chaffeensis* can be performed by using antibody research and/or molecular detection [7]. The latter is particularly effective in dogs with acute disease, but it loses sensitivity when dogs are in the asymptomatic phase. On the other hand, serological methods—especially when an immunofluorescent assay (IFA) is employed—need to be carefully used since crude antigens can compromise the specificity of an assay during epidemiological studies. Therefore, it is important to use specific antigens of *E. chaffeensis* to avoid false-positive results. In this sense, the TRP32 protein was previously molecularly characterized as being specific to *E. chaffeensis,* and anti-TRP32 antibodies did not cross-react with the closely related *E. canis* or other *Ehrlichia* species [8]. Therefore, the application of synthetic peptides in enzyme-linked immunosorbent assays (ELISA) is useful for epidemiological studies of *E. chaffeensis* and has been employed in serological surveys in the USA, Canada, and the Caribbean [9].

In Brazil, although serological evidence of human infection by this agent has been reported [10], studies concerning the occurrence of *E. chaffeensis* are scarce. As canines have a close relationship with humans, they are susceptible to different species of ticks and *E. chaffeensis*; therefore, they can be considered a bridge to human parasitism by ticks. Given the relevance of this issue, in this communication, we present the first serological evidence of *E. chaffeensis* in dogs from Brazil.

## 2. Materials and Methods

### 2.1. Samples

With reference to publications by Melo et al. [11,12] and Taques et al. [13,14], canine sera from a bank of samples created between 2011 and 2018 were analyzed. The serum samples covered all regions of Brazil: north (n = 124), northeast (n = 235), midwest (n = 450), southeast (123), and south (n = 202), totaling 1134 dogs. Except for those in the studies conducted by Melo et al. [11,12], where sampling was systematic, most samples were collected according to convenience from dogs that were treated at veterinary hospitals or received at public shelters, and they all had a history of tick parasitism. Except for the state of Amazonas, all states of Brazil were represented by the dog samples. All samples were tested against a crude *E. chaffeensis* antigen with an immunofluorescence assay (IFA) [2] and an *E. chaffeensis*-specific antigen (TRP32) with an enzyme-linked immunosorbent assay (ELISA) [8]. This study was approved by the Committee on Animal Research and Ethics of the Federal University of Mato Grosso (UFMT) under protocol no. 23108.122592/2015-10.

### 2.2. Immunofluorescence Assay (IFA)

An immunofluorescence assay (IFA) was used to detect antibodies on slides that were sensibilized with DH82 cells infected with the Arkansas strain of *E. chaffeensis* with a cut-off point at an initial dilution of 1:64 [2]. In each slide, sera from dogs previously used by Aguiar et al. [15] that were shown to be non-reactive (negative control) and a known reactive serum (positive control) were included.

### 2.3. TRP32-Enzyme-Linked Immunosorbent Assay (ELISA)

Peptides corresponding to the immunodominant repeat regions of TRP32 from *E. chaffeensis* (23-mer, SDLHGSFSVELFDPFKEAVQLGN) [8] were synthesized (Bio-Synthesis Inc., Lewisville, TX) and employed as antigens in ELISA. A peptide corresponding to the C-terminal region of TRP36 from the Israeli strain of *E. canis* (IS36-C-V, 15-mer, NPTGLKFLDLYTQLTL) [13] was used as a negative peptide control. All peptides (lyophilized) were resuspended in molecular-grade water at 1 mg/mL. ELISA plates (MaxiSorp; Nunc, Roskilde, Denmark) were coated (1.0 µg/well) with the corresponding synthetic peptides suspended in phosphate-buffered saline (pH 7.4), and the assay was performed as previously described [8]. The development of color was determined with a microplate reader (Epoch™; Biotek Instruments, Winooski, VT, USA), and the data were analyzed using the Gen5™ data analysis software (Biotek Instruments, USA). The optical density (OD; A650) readings represent the mean for three wells (±standard deviations) after subtracting the OD value of the non-reactive control peptide (IS36-C-V). All dog sera that were negative for *E. chaffeensis* infection in IFA had mean OD readings of <0.050; therefore, a positive sample threshold was set at >0.150 OD units above the negative control absorbance.

### 2.4. Statistical Analysis

The differences between the results found in each region and positivity for *E. chaffeensis* antibodies were evaluated by using the Chi-square test or exact Fisher test, and *p* ≤ 0.05 was considered significant. The statistical analysis was performed using Epi Info^TM^ version 5.5.1.

## 3. Results

The serological analyses showed that there were nine (0.7%) seropositive dogs according to the ELISA assay, and seven (0.6%) of them had IFA antibody titers ranging from 160 to 5120. All regions of Brazil had at least one reactive dog (Figure 1). Similar frequencies were observed among dogs that were seropositive for *E. chaffeensis* in all Brazilian regions, so there were no statistically significant differences (*p* > 0.05). The identification, antibody titers, and OD values for each sample and region are shown in Table 1.

## 4. Discussion

In this survey, we used two different types of antigens to assess the presence of antibodies against *E. chaffeensis*: DH82 cells infected with an Arkansas isolate were used for an indirect immunofluorescence reaction [2], and synthetic peptides of the TRP32 protein were used for ELISA [8]. Although the first technique presents high sensitivity, it lacks specificity, as the crude antigens of *E. chaffeensis* may share similar epitopes with *E. canis* or *Ehrlichia minasensis*, which are other members of the *Ehrlichia* genus that are endemic among dogs (*E. canis*) [14] or present in dogs according to serological detection (*E. minasensis*) [16] in Brazil. On the other hand, the peptides of the TRP32 protein from *E. chaffeensis* are characterized by their high specificity; moreover, the TRP32-ELISA also guaranteed an appropriate sensitivity when detecting samples in the context of seroconversion, even in comparison with the IFA technique, as can be seen in two of the samples shown in Table 1 (#9 and #307), where antibodies were detected by ELISA but did not have detectable titers in the IFA. This result can be expected with the use of this antigen in an ELISA platform since a similar result was observed in a previous study when comparing TRP19/36 protein antigens by using IFA during the initial stage of infection by *E. canis* [17].

Although the specificity of the TRP32 peptide was supported by a previous experiment [8], we preferred to consider that the antibodies detected in this study could not only be stimulated by *E. chaffeensis* infection but could also come from infection by a closely related species. The South American isolates should be more extensively investigated, especially for other genetic targets, as the detections in most studies have been based on one or two genes (16S rRNA, *dsb*) [3,4,5,6]. Hence, our results reinforce the evidence of the occurrence of *E. chaffeensis* or a closely related species in South America, and this is the first survey to evaluate the occurrence of this pathogen in dogs in Brazil through the detection of antibodies.

The samples that were tested came from all states of Brazil except for the state of Amazonas; i.e., the survey was comprehensive and involved all Brazilian ecosystems and biomes. In contrast to the findings on the prevalence of anti-*E. canis* antibodies detected in dogs in Brazil, with rates ranging from 16 to 57% [14], the frequency of antibodies detected in the present study (0.7%) for *E. chaffeensis* demonstrated a low occurrence of this agent among dogs. Beall et al. [18] evaluated dogs from several states of the USA and estimated that the prevalence of anti-*E. chaffeensis* antibodies ranged from 0.1 to 21.4% depending on the location evaluated and the presence of *A. americanum*.

The presence of seropositive dogs from all regions of Brazil also suggests that the occurrence of this species is not restricted to the Pantanal, other flooded Brazilian regions, or even the Southern Cone of South America (Argentina, Brazil, and Uruguay), as these are the regions in which the presence of marsh deer (*B. dichotomus*), one of the probable hosts of this species of *Ehrlichia,* has been reported [5]. Nevertheless, even with no statistical significance, we observed that three (30%) of the nine seropositive samples were from dogs from the municipality of Poconé, located in the Brazilian Pantanal, which is a great environment for animal diversity in Brazil. Unfortunately, the sample data were limited from an epidemiological point of view, since we did not have access to more information on most of the tested animals. For example, only in the survey by Melo et al. [11,12] were we able to distinguish which animals were from urban (Table 1; U36 and U50) and rural (R157) areas; some other samples came from large cities, such as São Paulo, Curitiba, and Porto Alegre, while others came from smaller cities in the states.

Although our results reinforce the hypothesis of the presence of *E. chaffeensis* in Brazil, the seropositivity found in dogs from different Brazilian regions supports the debate on the probable tick species involved in transmission. *Ehrlichia* spp. that are closely related to *E. chaffeensis* have been detected in *A. parvum* [3], *A. tigrinum* [4], and *R. microplus* [5] ticks from Argentina. Despite occurring in a wide territorial range (Mexico to Argentina), *Amblyomma parvum* is primarily found in rural areas [19], and the serum bank that was tested here predominantly contained samples from dogs in urban areas [9,10,11,12]. On the other hand, *A. tigrinum* has restricted areas of occurrence (the Southern Cone of South America) [19]. Therefore, the detection of anti-*E. chaffeensis* antibodies in dogs from regions in which this species does not occur would not be justified. *Rhipicephalus microplus* is predominantly distributed in tropical and subtropical areas worldwide and is strongly associated with cattle. Nevertheless, there are records of *R. microplus* in various hosts, from other mammals and birds to amphibians [19]. However, it is important to emphasize that in this tick, the presence of DNA compatible with *E. chaffeensis* was found in specimens that parasitized *B. dichotomus* [5].

Interestingly, *E. chaffeensis* was previously detected in *Rhipicephalus sanguineus* sensu lato ticks in Cameroon, where *A. americanum* is not found [20,21]. Additionally, the transmission of *E. chaffeensis* to dogs by *R. sanguineus* s. l. was experimentally observed [22]. Therefore, considering the widespread distribution of *R. sanguineus* s. l. in Brazil, this tick must be intensely investigated for other species of *Ehrlichia* in addition to *E. canis*.

## 5. Conclusions

We conclude that dogs from different areas of Brazil were exposed to *E. chaffeensis* or a closely related species. Due to the different characteristics of the states and their regions, the probable transmitting ticks of these species of *Ehrlichia* must be further investigated. The data presented in this communication reinforces the need to discuss tick control strategies, as these parasites, which are deeply involved in animal health, can transmit important pathogens to humans. Due to the disorders caused by *E. chaffeensis* in the United States of America and developing countries, such as Brazil, which has several areas with sanitary and environmental deficits, infections transmitted by ticks can have negative impacts on public health.

## Figures and Tables

**Figure 1 pathogens-12-01024-f001:**
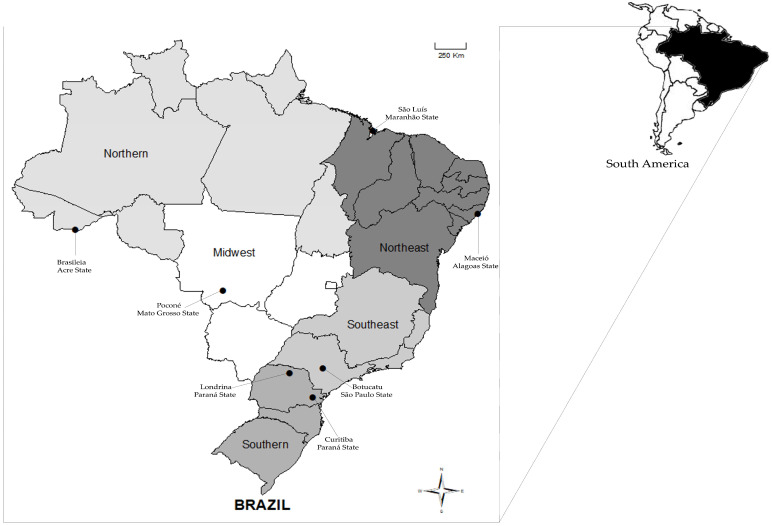
Geographic distribution (northern, midwestern, northeastern, southern, and southeastern regions) of the serological detection of *Ehrlichia chaffeensis* antibodies in dogs from Brazil.

**Table 1 pathogens-12-01024-t001:** Sample identification and origin of the dog samples, optical density values for TRP32-ELISA, and titers of anti-*Ehrlichia chaffeensis* antibodies according to IFA.

Identification	Municipality/State	TRP32-ELISA	*E. chaffeensis* IFA
#9	Brasileia/Acre	0.331	NR *
U36	Poconé/Mato Grosso	0.150	2560
U50	Poconé/Mato Grosso	0.623	160
R157	Poconé/Mato Grosso	0.526	5120
#6	São Luís/Maranhão	0.493	1280
#5	Maceió/Alagoas	1.242	2560
#8	Botucatu/São Paulo	0.516	2560
#4	Londrina/Paraná	0.363	1280
#307	Curitiba/Paraná	0.716	NR *

* NR means not reagent.

## Data Availability

Data sharing is not applicable.

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
