# Peer review of "Serological Evidence Supporting the Occurrence of *Ehrlichia chaffeensis* or a Closely Related Species in Brazilian Dogs"

_pathogens, 2023, doi:10.3390/pathogens12081024_

Round 1
Reviewer 1 Report
Dear authors,
congratulations for this very well written short communication. The study is properly designed and the number of samples is adequate for the aim of the study. The results are clearly presented and well discussed. The knowledge of the presence of E. chaffeensis antibodies in dogs from all parts of Brazil is crucial for further research according to this pathogenic bacterium and for the development of possible prevention strategies by public health facilities. I just found some minor spelling and format errors that are listed below:
line 37: please use Ehrlichia sp. instead of Rickettsiae
line 41: please use Ehrlichia sp. instead of rickettsia species
lines 52-79: please use line justification
line 71: Denmark instead of Denamark
line 81: please add % for the seven IFA positive samples
line 83: please use OD, as this abbreviation was introduced earlier (see line 75)
line 95: E. canis instead of Ehrlichia canis as E. canis was named earlier in the text
line 108: 16S rRNA not in italic, the gens´ name is rrs
line 123: Ehrlichia spp. instead of ehrlichia
Author Response
Reviewer 1
Dear authors,
Congratulations for this very well written short communication. The study is properly designed and the number of samples is adequate for the aim of the study. The results are clearly presented and well discussed. The knowledge of the presence of E. chaffeensis antibodies in dogs from all parts of Brazil is crucial for further research according to this pathogenic bacterium and for the development of possible prevention strategies by public health facilities. I just found some minor spelling and format errors that are listed below:
-line 37: please use Ehrlichia sp. instead of Rickettsiae
Author response: The term has been replaced as requested.
-line 41: please use Ehrlichia sp. instead of rickettsia species
Author response: The term has been replaced as requested.
-lines 52-79: please use line justification
Author response: The line was justificated.
-line 71: Denmark instead of Denamark
Author response: The term has been replaced as requested.
-line 81: please add % for the seven IFA positive samples
Author response: The % was added.
-line 83: please use OD, as this abbreviation was introduced earlier (see line 75)
Author response: The OD abbreviation was used.
-line 95: E. canis instead of Ehrlichia canis as E. canis was named earlier in the text
Author response: The abbreviation was used.
-line 108: 16S rRNA not in italic, the gens´ name is rrs
Author response: The italic form was removed.
-line 123: Ehrlichia spp. instead of ehrlichia
Author response: The term has been replaced as requested.
Reviewer 2 Report
The study titled "Serological evidence of Ehrlichia chaffeensis in Brazilian dogs" sought to examine serological evidence of E. chaffeensis or a closely related species in dogs from Brazil. To achieve this, the authors employed crude antigens of E. chaffeensis in an indirect immunofluorescence reaction and utilized synthetic peptides of the TRP32 protein in an ELISA assay. Understanding the prevalence and distribution of this bacterium among dogs’ populations helps in tracking disease patterns and developing effective control strategies. Besides the novelty of the topic, the manuscript contains very limited information and lacks effective organization with regards to various critical elements such as data presentation, and technical write-up. Moreover, upon reviewing the Abstract, Results, and Discussion sections, it appears that the results were not presented based on statistical analysis. Merely discussing the results based on percentage values, without taking into account the statistical analysis, lacks scientific rigor and will not provide a meaningful contribution to the scientific community.
Additionally, the study lacks a conclusion section at the end, which is essential for summarizing the key findings and implications of the research.
Upon conducting a comprehensive review of the manuscript, I have encountered noteworthy concerns pertaining to the article.
Overall, English quality is fine.
Author Response
The study titled "Serological evidence of Ehrlichia chaffeensis in Brazilian dogs" sought to examine serological evidence of E. chaffeensis or a closely related species in dogs from Brazil. To achieve this, the authors employed crude antigens of E. chaffeensis in an indirect immunofluorescence reaction and utilized synthetic peptides of the TRP32 protein in an ELISA assay. Understanding the prevalence and distribution of this bacterium among dogs’ populations helps in tracking disease patterns and developing effective control strategies. Besides the novelty of the topic, the manuscript contains very limited information and lacks effective organization with regards to various critical elements such as data presentation, and technical write-up. Moreover, upon reviewing the Abstract, Results, and Discussion sections, it appears that the results were not presented based on statistical analysis. Merely discussing the results based on percentage values, without taking into account the statistical analysis, lacks scientific rigor and will not provide a meaningful contribution to the scientific community.
Additionally, the study lacks a conclusion section at the end, which is essential for summarizing the key findings and implications of the research.
Upon conducting a comprehensive review of the manuscript, I have encountered noteworthy concerns pertaining to the article.
Author response:
We tried to carry out this survey with the better efficient serological method for anti-E. chaffeensis detection in Brazil. In addition, we have an extensive bank of canine serum, consisting of samples from several states in Brazil. As there is no - according to our understanding about canine ehrlichiosis caused by E. chaffeensis - any serological survey in Brazil, or even any investigation about this species of ehrlichia in dogs, we were led to explore this issue that expands the knowledge about the occurrence of this agent beyond the United States.
The points addressed by this reviewer were commented on by the other reviewers, many underwent adjustments, and therefore we believe that our manuscript is able to be accepted.
We had not really performed a statistical analysis of the results because our objective was to communicate the presence of anti-E. chaffeensis antibodies in dogs from Brazil and mainly because the seropositive rate was low. Even so, we performed the statistical analysis and, as expected, no significant association was detected.
We do not agree with the judgment that our manuscript is not organized, as we seek to meet the necessary requirements for the scientific foundation and its hypotheses, we seek to make clear the origin of the samples as well as the methodology used. We are aware of the limitations of our study, mainly because it is often difficult to obtain a national sample bank with all available data, but we make this clear in the discussion about this study limitation. Still, our results demonstrated what we were looking for; there are dogs in Brazil being exposed to E. chaffeensis or some close-related species.
Finally, I agree with the conclusion. Therefore, include after the discussion the conclusion below:
"5. Conclusion
We conclude that dogs from different areas in Brazil were exposed to E. chaffeensis or a close-related species. Due to the different characteristics of the states and their regions, the probable transmitting tick of this Ehrlichia must be further investigated. The data presented in this communication reinforce the need to discuss tick control strategies, as these parasites, which are so involved in animal health, can transmit important pathogens to humans. Considering the disorders caused by E. chaffeensis in the United States of America; developing countries, such as Brazil, with several areas with sanitary and environmental deficit, infections transmitted by ticks can cause negative impacts on public health."
Reviewer 3 Report
The manuscript entitled "Serological evidence of Ehrlichia chaffeensis in Brazilian dogs" was reviewed. This is a very interesting study providing insight into an issue with potential concern for Public Health.
A word is missing in the title of the manuscript. Serological evidence of Ehrlichia chaffensis what exposure, infection, antibody circulation? Please add the word or rephrase the title.
Regarding methodology, the way the negative control for IFAT was selected should be clarified.
The presentation of results especially regarding seroprevalence in different areas would benefit from a map pointing out the areas. Moreover, the difference in seropositivity in different areas could be discussed although the low number of positive dogs cannot allow for a statistical analysis.
The two different methods had different sensitivity and specificity based on the antigen used and the possibility misidentify samples due to cross-reactions as the authors have pointed out. However, when the more specific method (ELISA) was used two samples were found to be positive but negative in IFAT. The authors should comment on this finding.
Minor linguistic problems have been detected mainly word misspelling. Most of the text is fine but splitting some sentences could make them more easy to follow.
Author Response
Reviewer 3
The manuscript entitled "Serological evidence of Ehrlichia chaffeensis in Brazilian dogs" was reviewed. This is a very interesting study providing insight into an issue with potential concern for Public Health.
-A word is missing in the title of the manuscript. Serological evidence of Ehrlichia chaffensis what exposure, infection, antibody circulation? Please add the word or rephrase the title.
Author response: Serological evidence supports the occurrence of Ehrlichia chaffeensis or a close species in Brazilian dogs.
-Regarding methodology, the way the negative control for IFAT was selected should be clarified.
Author response: The source of the negative and positive controls used in the experiment was indicated. “In each slide, a serum of dogs previously used by Aguiar et al. [15] shown to be non-reactive (negative control) and a known reactive serum (positive control) were included”.
-The presentation of results especially regarding seroprevalence in different areas would benefit from a map pointing out the areas. Moreover, the difference in seropositivity in different areas could be discussed although the low number of positive dogs cannot allow for a statistical analysis.
Author response: The suggested map was built and is shown in figure 1. In relation to the number of positive sera and statistical analysis, we included in the material and methods the statistical analysis (between lines 100-103), in the results, the non-significance finding (between line 107-108) and have included a sentence in the discussion between lines 153-156.
Line 100-103: Differences between the results found per region and positivity of E. chaffeensis antibodies were evaluated by the chi-square test or Exact of Fisher and p ≤ 0.05 was considered significant. The statistical analysis was performed using Epi InfoTM version 5.5.1 software.
Line 107-108: Similar results were observed among the dogs reactive to E. chaffeensis in all the Brazilian regions (p > 0.05).
Line 153-156: Nevertheless, even with no statistical significance, we observed that three (30%) of the nine seropositive samples were from dogs from the municipality of Poconé, located in the Brazilian Pantanal, a great environment for animal diversity in Brazil.
-The two different methods had different sensitivity and specificity based on the antigen used and the possibility misidentify samples due to cross-reactions as the authors have pointed out. However, when the more specific method (ELISA) was used two samples were found to be positive but negative in IFAT. The authors should comment on this finding.
Author response: This result was discussed between lines 124-128 as follows:
“On the other hand, the peptides of the TRP32 protein from E. chaffeensis is characterized by its high specificity, moreover the TRP32 ELISA also guarantees appropriate sensitivity to detect samples in seroconversion, even when compared to the IFA technique, as can be seen in two samples shown in Table 1 (#9 and #307), where antibodies were detected by ELISA but without detectable titers in IFA. This result can be expected with the use of this antigen in an ELISA platform, since a similar result was observed in a previous study when comparing TRP19/36 protein antigens with IFA during the initial stage of infection by E. canis [17].”
Comments on the Quality of English Language
Minor linguistic problems have been detected mainly word misspelling. Most of the text is fine but splitting some sentences could make them more easy to follow.
Reviewer 4 Report
The work titled "Serological evidence of Ehrlichia chaffeensis in Brazilian dogs " and performed by Taques et al., is a seroprevalence study for E.chaffensis performed in the dog in Brazil. The authors tested a large number of samples with two serological tests. However, there are some methodological errors. In fact, the materials and methods section should be deepened with more information on the sampling and tests used (in some cases the information is provided only in the discussion section). The authors also do not properly discuss their data.
Line 39: This sentence is confusing, please rephrase it.
Line 43: I am not aware that information about E. chaffensis is limited in South America. Authors may delete these references or rephrase them.
Line 46: What do the authors mean by “close related species”? Please, specify.
Line 48-51: This information should not be included in the introduction. They should be explained in materials and methods.
Line 53: It is fine to include a reference for sampling, but authors should provide more information about sampling.
Material and methods: The authors are asked to provide more information on the ELISA and the IFA performed. The authors may also consider merging the two subsections related to serological testing. They should also justify their use.
Line 71: I think it's a type error. 1 gram/well?
Line 81: Please delete “assay” (it is already included in ELISA: Enzyme linked immunorbent assay).
Lines 94-96: This part is misleading.
Lines 99-102: How are these sentences supported by the facts? I advise the authors not to refer to individual sera.
The discussion is very problematic. The authors talk about the performances of the tests used (these things should be explained in materials and methods to explain why certain tests have been chosen as a reference) while they do not compare their data with those obtained in other countries, in other species, or for other species of Ehrlichia (in this case I recommend some references to the authors to get ideas for a few sentences of the discussion). doi: 10.3390/ani11010009.
Line 111: “seroreagent dogs”. Please, use specific and appropriate terms.
Line 123: Please, change “ehrlichia” in “Ehrlichia”.
Some parts of the manuscript are confused. Some terms used inappropriately. Authors could improve the level of the manuscript without using the editing service at this stage.
Author Response
Reviewer 4
The work titled "Serological evidence of Ehrlichia chaffeensis in Brazilian dogs " and performed by Taques et al., is a seroprevalence study for E. chaffensis performed in the dog in Brazil. The authors tested a large number of samples with two serological tests. However, there are some methodological errors. In fact, the materials and methods section should be deepened with more information on the sampling and tests used (in some cases the information is provided only in the discussion section). The authors also do not properly discuss their data.
-Line 39: This sentence is confusing, please rephrase it.
Author response: Thank you for bringing this sentence to our attention. We change to the following: “In South America, specifically Argentina and Brazil, Ehrlichia sp. similar or close related to E. chaffeensis have been detected in the ticks Amblyomma parvum [3], Amblyomma tigrinum [4], Rhipicephalus microplus and in marsh deer (Blastocerus dichotomus) [5] and non-human primate (Mico melanurus) [6].”
-Line 43: I am not aware that information about E. chaffeensis is limited in South America. Authors may delete these references or rephrase them.
Author response: This question was not very clear. However, in this sentence we tried to report the findings of E. chaffeensis or a close species in Brazil and Argentina. There are other reports of E. chaffeensis in South America, but those reported in the introduction are the most robust and served the justice of our investigation.
-Line 46: What do the authors mean by “close related species”? Please, specify.
Author response: We understand that this sentence has been confusing and have decided to remove the term cited by this reviewer.
-Line 48-51: This information should not be included in the introduction. They should be explained in materials and methods.
Author response: Thank you for drawing our attention to this paragraph, but after reflection, we chose to add information in the introduction (line44-55) about the diagnosis of E. chaffeensis infection in dogs, and not add this information in the M&M, where in my experience the methodology of reactions should be described. I therefore believe that it is not necessary to unify the serological tests in a single subsection.
-Line 53: It is fine to include a reference for sampling, but authors should provide more information about sampling.
Author response: Thank you for this guidance. The following information was added in the material and methods on line 68-72. “Except for the studies conducted by Melo et al. [11], [12] where sampling was systematic, most samples were collected for convenience, from dogs that were treated at veterinary hospitals or received at public shelters, all with a history of tick parasitism. Except for the state of Amazonas, all other states in Brazil were contemplated with dog samples.”
-Material and methods: The authors are asked to provide more information on the ELISA and the IFA performed. The authors may also consider merging the two subsections related to serological testing. They should also justify their use.
Author response:
Thank you for drawing our attention to this paragraph, but after reflection, we chose to add information in the introduction (line 44) about the diagnosis of E. chaffeensis infection in dogs, and not add this information in the M&M, where in my experience the methodology of reactions should be described. I therefore believe that it is not necessary to unify the serological tests in a single subsection.
-Line 71: I think it's a type error. 1 gram/well?
Author response: Thank you. It was a type error. The correct unit is µg/well. It was corrected.
-Line 81: Please delete “assay” (it is already included in ELISA: Enzyme linked immunorbent assay).
Author response: The “assay” term was deleted.
-Lines 94-96: This part is misleading.
Author response: There seems to be a problem with the line count in the document. I couldn't identify what happened. But I transferred the paragraph referring to the ELISA to the next page and there was no change in the text.
-Lines 99-102: How are these sentences supported by the facts? I advise the authors not to refer to individual sera.
Author response: I apologize, but the question was not clear to me.
-The discussion is very problematic. The authors talk about the performances of the tests used (these things should be explained in materials and methods to explain why certain tests have been chosen as a reference) while they do not compare their data with those obtained in other countries, in other species, or for other species of Ehrlichia (in this case I recommend some references to the authors to get ideas for a few sentences of the discussion). doi: 10.3390/ani11010009.
Author response: To meet this suggestion, we add the following text between lines 141 and 148: “Except for the state of Amazonas, the samples tested came from all states in Brazil, that is, the survey is comprehensive and involved all Brazilian ecosystems and biomes. Contrary to the prevalence of anti-E. canis in dogs detected in Brazil, with rates ranging from 16 to 57% [14], the frequency of antibodies detected in the present study (0.7%) for E. chaffeensis demonstrated low occurrence of the agent among dogs. Beall et al. [18], evaluated dogs from several states from the USA, and estimated the prevalence of anti-E. chaffeensis ranging from 0.1 to 21.4% depending on the location evaluated and the presence of A. americanum.”
Line 111: “seroreagent dogs”. Please, use specific and appropriate terms.
Author response: The word was changed to seropositive.
Line 123: Please, change “ehrlichia” in “Ehrlichia”.
Author response: The word was changed.
Comments on the Quality of English Language
Some parts of the manuscript are confused. Some terms used inappropriately. Authors could improve the level of the manuscript without using the editing service at this stage.
Round 2
Reviewer 2 Report
I highly appreciate the authors' diligent endeavors in enhancing the manuscript's quality. The considerable improvements made have substantially bolstered the scientific validity of the article. As a result, I am confident in endorsing the acceptance of the manuscript, considering the revisions implemented.
Some minor revisions to the English language are needed.
Author Response
I am immensely grateful to reviewer #2 for the suggestions, corrections and contributions to the improvement of the manuscript.
Reviewer 4 Report
I thank the authors for directing most of my comments; I see that the manuscript has increased in quality and readability. In my opinion, the manuscript is almost ready to be accepted; it needs some moderate corrections, which I list below.
Abstract:
Line 22: "immunofluorescence" could be "IFA" as written in Line 24. Please delete "assay" after "ELISA" (which is already an acronym for enzyme-linked immunorbent ASSAY).
Line 24: Authors should consider writing of antibody titers rather than "IFA antibodies".
Line 25: Please change "reactive dog" to "seropositive animals" or something similar. The sample was reactive, not the dog.
Line 27: In addition to being a bridge for human parasitism, the dog could also represent an environmental sentinel for these infections (both things are yet to be proven). Authors could emphasize these aspects in the abstract and throughout the manuscript.
Introduction:
Lines 42-43: This sentence is not entirely correct and can appear foregone (if we consider that man is a primate). "Primate" should be changed to "non-human primate".
Line 59: As abstract section.
Results:
The statistical analysis that the authors describe in the materials and methods section does not appear in the results. The authors may include a table relating to this analysis in the manuscript. Also, the results section is a bit sparse. Did the authors consider comparing the two ELISAs used and commenting on the degree of agreement between the two tests in relation to the IFA?
I advise authors to check the numbering of each reference.
The quality of English has improved significantly since the last revision and may be sufficient.
Author Response
I thank the authors for directing most of my comments; I see that the manuscript has increased in quality and readability. In my opinion, the manuscript is almost ready to be accepted; it needs some moderate corrections, which I list below.
Abstract:
Line 22: "immunofluorescence" could be "IFA" as written in Line 24. Please delete "assay" after "ELISA" (which is already an acronym for enzyme-linked immunorbent ASSAY).
Author response: Thanks for the suggestion and correction. Both accepted.
Line 24: Authors should consider writing of antibody titers rather than "IFA antibodies".
Author response: Thanks for the correction. The term antibodies has been replaced by titers.
Line 25: Please change "reactive dog" to "seropositive animals" or something similar. The sample was reactive, not the dog.
Author response: Thanks for the correction. The sentence was as follows: “All regions of Brazil presented at least one seropositive dog.”
Line 27: In addition to being a bridge for human parasitism, the dog could also represent an environmental sentinel for these infections (both things are yet to be proven). Authors could emphasize these aspects in the abstract and throughout the manuscript.
Author response: Thanks for the suggestion. The sentence was as follows: “. As dogs have a close relationship with humans, they can be used as an environmental sentinel for these infections because they can act as a bridge to human parasitism or infection with ehrlichial agents.” We took the opportunity to include the following sentence between lines 38 and 39: “for this reason, they can be considered environmental sentinels for this agent.”
Introduction:
Lines 42-43: This sentence is not entirely correct and can appear foregone (if we consider that man is a primate). "Primate" should be changed to "non-human primate".
Author response: Thanks for the correction. The term non-human primate is updated in the sentence.
Line 59: As abstract section.
Author response: I'm sorry but I didn't understand what was flagged.
Results:
The statistical analysis that the authors describe in the materials and methods section does not appear in the results. The authors may include a table relating to this analysis in the manuscript. Also, the results section is a bit sparse. Did the authors consider comparing the two ELISAs used and commenting on the degree of agreement between the two tests in relation to the IFA?
Author response: Thank you for drawing attention to the results. We adjust the results in line 116 to emphasize the result of the statistical analysis. “Similar frequencies were observed among dogs that were seropositive for E. chaffeensis in all Brazilian regions, so there were no statistically significant differences (p > 0.05).” We used only one kind of ELISA and we did not evaluate the concordance between the diagnostic tests, as the result would be in our opinion a little speculative since there is no consensus on a true gold standard test for this agent. Anyway, thank you for this question.
I advise authors to check the numbering of each reference.
Author response: All references were checked.
Line 111: “seroreagent dogs”. Please, use specific and appropriate terms.
Author response: The word was changed to seropositive.
Line 123: Please, change “ehrlichia” in “Ehrlichia”.
Author response: The word was changed.
Comments on the Quality of English Language
Some parts of the manuscript are confused. Some terms used inappropriately. Authors could improve the level of the manuscript without using the editing service at this stage.
Author response: The manuscript underwent grammatical review.